# Effects of Sex Class, a Combined Androgen and Estrogen Implant, and Pasture Supplementation on Growth and Carcass Performance and Meat Quality of Zebu-Type Grass-Fed Cattle

**DOI:** 10.3390/ani11123441

**Published:** 2021-12-02

**Authors:** Nelson Huerta-Leidenz, Nancy Jerez-Timaure, Jhones Onorino Sarturi, Mindy M. Brashears, Markus F. Miller, Alexis Moya, Susmira Godoy

**Affiliations:** 1International Center for Food Industry Excellence, Department of Animal and Food Sciences, Texas Tech University, P.O. Box 42141, Lubbock, TX 79409, USA; j.sarturi@ttu.edu (J.O.S.); Mindy.Brashears@ttu.edu (M.M.B.); mfmrraider@aol.com (M.F.M.); 2Instituto de Ciencia Animal, Facultad de Ciencias Veterinarias, Universidad Austral de Chile, Valdivia 5090000, Chile; nancy.jerez@uach.cl; 3MOYAMIX C.A. Ave Mariño Sur, Centro Empresarial UniAragua, Maracay 2103, Aragua, Venezuela; moyaalexis69@gmail.com; 4Instituto Nacional de Investigaciones Agrícolas, Centro Nacional de Investigaciones Agropecuarias, Maracay 2101, Aragua, Venezuela; susmigodoy@gmail.com

**Keywords:** growth rate, carcass yield, beef quality, tenderness, trenbolone acetate, sex class

## Abstract

**Simple Summary:**

Inadequate feeding and management practices of cattle breeders in the Apure’s savanna (Venezuela) impede the adoption of cow–calf-to-finish systems on their own ranches. This study evaluated the feasibility of finishing Zebu-type young cattle on savanna cultivated pastures with better growth and carcass traits and meat quality. Castration can be performed without detrimental consequences for cattle growth, eliciting a desirable response in live weight gain to the implant, while improving carcass finish, the yield of valuable cuts, and meat tenderness. The implant enhanced the rate of gain of steers only, with no beneficial effects on cut-out yields or adverse impacts on meat quality. The supplementation improved key performance traits, allowing cattle to be harvested at a younger age, which is clearly advantageous for rotating more animals per production cycle. Moreover, carcass quality was marginally improved, but tenderness was negatively affected. The data presented herein may be used as a benchmark for producers of similar cattle in tropical ecosystems throughout the world.

**Abstract:**

Forty-seven Zebu calves were used to determine the effects of class (bull or steer), supplementation (SUPPL, a poultry litter-based supplement or mineral supplementation), and implant (20 mg estradiol combined with 120 mg of trenbolone acetate or no implant) on growth and carcass performance and beef eating quality. The average daily gain (ADG) of implanted cattle significantly increased for steers, but not for bulls. The SUPPL treatment increased ADG by 8.63% from day 0 to end, and shortened in 73.3 d the time to reach 480 kg BW (*p* < 0.01). Compared to bulls, the steer carcasses exhibited more desirable maturity and finish scores, thicker back fat (*p* < 0.05), and yielded greater (*p* < 0.01) percentages of high-value boneless subprimals (HVBLS) (+1.64%) and total cuts (1.35%). The SUPPL bulls dressed 2.63 and 1.63% greater than non-supplemented bulls and SUPPL steers, respectively (*p* < 0.05). Meat sensory quality was subtly affected (*p* < 0.05) by sex class or supplementation. The implant did not affect (*p* > 0.05) shear force or sensory ratings. The supplementation improved key growth performance traits while it adversely affected tenderness-related sensory traits. The implant enhanced the rate of gain of steers only, without improving cut-out yields or inducing adverse effects on palatability traits in both steers and bulls.

## 1. Introduction

Due to the harsh environmental conditions in neotropical savannas of South America, the profitable production of grass-fed Zebu-type cattle faces multiple challenges. Hence, most of the ranches are exclusively devoted to extensive cow–calf operations with low inputs. This is not necessarily the situation of the grazing lowlands of Apure State, Venezuela, where a prominent hydraulic infrastructure has allowed flood control during the wet season, water supply for cattle and pasture irrigation during the dry season [1,2], and the introduction of new forage species, with better nutritional quality than the native savanna vegetation [3]. Under these improved savanna conditions, diligent ranchers are committed to adopting a grass-based beef cow–calf-to-finishing system to increase productivity. A feasible, three-way strategy to improve production efficiency and beef quality involves the concomitant use of castration due to an increase in meat sensory attributes, hormonal growth promoters due to an extension of the growth curve, and pasture supplementation to ensure that proper nutrient requirements are met. Castration is not a generalized practice in Tropical America due to alleged decreases in growth rate and feed efficiency compared to intact, non-implanted counterparts [4,5,6]. However, detrimental effects of castration on growth traits and cutability have not been noted in grass-fed, tropically adapted cattle [4,7]. Despite the long-standing use of hormone growth promoters (HGP) to counteract the hindering castration effects on cattle growth performance [8,9], there is a lack of reports on the response of tropical-grazing cattle to potent hormonal implants (i.e., the combination of estradiol benzoate and trenbolone acetate). It has also been acknowledged that the implant’s effectiveness will also depend upon the animal’s plane of nutrition [10]. Because of the marked seasonality and variation in the availability and quality of pastures, the supplementation of key nutrients is necessary to maintain production and avoid the costs of carrying the slaughter cattle through additional dry seasons to meet the desired endpoints [11]. Poultry litter-based supplementation has been shown to be beneficial in avoiding BW loss or improving growth rate [12,13], but its effects on carcass characteristics and/or beef palatability are still under scrutiny [14,15]. We hypothesize that the use of non-protein nitrogen energy pasture supplementation might elicit greater responses to steroidal implants in beef quantity and/or quality from grazing cattle, particularly from castrated males. Therefore, the aim of this experiment was to determine the effects of castration (sex class), steroidal hormone implantation, strategic supplementation, and potential interactions on growth traits, carcass performance, and eating quality of grass-fed Brahman-influenced cattle.

## 2. Materials and Methods

This study was carried out in compliance with the guidelines of the bioethics code for animal experiments of the Venezuelan National Council for Scientific and Technological Research (FONACIT) [16] adopted by the Institute of Agronomical Research and overseen by the Council for Scientific and Humanistic Development at La Universidad del Zulia (CONDES-LUZ). Project Protocol CONDES-LUZ # CC-0390-04. 

### 2.1. Location, Animal Handling, and Experiment Conditions

The experiment was carried out on a commercial ranch located in the Modules of Apure’s savanna of (southwestern Venezuelan Llanos). Previous reports have described the agroecological characteristics of the ecosystem [1,15,17]. Homogenous single-sourced (*n* = 47; Mean BW ± SEM = 339.02 ± 27.8 kg) Brahman-influenced male calves were used for the current experiment. Predominantly crossbreed Zebu calves came from a herd of cows with a high proportion of Zebu mated with purebred Brahman sires. During the pre-weaning stage, the calves stayed with dams on a good quality, native-range pasture with access to minerals to meet or exceed mineral nutrient requirements (NASEM) [18]. Calves to be castrated were handled as not to suffer unusual stress and pain as mandated by policies of Fondo Nacional de Ciencia, Tecnología e Innovación [16]. Briefly, calves were restrained into a squeeze chute and surgical castration was quickly performed by an experienced veterinarian. Prior to the surgical procedure, testicles and the surrounding anatomical zone were disinfected by washing with a fluid soap containing povidone–iodine 5% solution (Betadine® Avrio Health L.P, New York, NY, USA). Following the removal of testicles from scrotal bags, a dry powder antiseptic spray (Betadine® dry powder, Avrio Health L.P, USA) was applied into the scrotal bags, which were later sprayed with an antiseptic ointment (Bactrovec plata AM®, König, Argentina). Castrated (steers) and intact (bulls) calves were treated against ectoparasites and endoparasites with ivermectin (Ivermetopp Dorado, Topp Laboratories, Caracas, Venezuela) subcutaneously injected at the recommended dose of 1 mL/50 Kg BW, and vaccinated for foot-and-mouth disease (Aftovac®, CALA Laboratories, Maracay, Venezuela). The implanted group received a single dose of a synthetic steroidal implant containing 20 mg estradiol benzoate + 140 mg trenbolone acetate (EB + TBA) (Revalor® Hoechst-Roussel Agri-Vet, Somerville, NJ, USA; international dosage) immediately before they entered the selected grazing area. Paddocks consisted of cultivated grasslands (*Brachiaria radicans*). The supplementation experiment was carried out at the start of the dry season. The control group was kept in these pastures under a rotational system with grazing periods of 32 days (4 days of occupation and 28 days of rest) at a stocking rate of 1.1 AU/ha (AU = 450 kg), with free access to a mineral mix (10% P, 16% Ca, 13% Na, 1% S, 1% Mg, 0.5% Zn, 0.2% Cu, 0.008% I, 0.002% Co, 0.002% Se), whose daily consumption averaged 80 g/animal throughout the experiment. The cultivated pasture contained on average (DM basis) 63% TDN, 6% CP, 47% nitrogen-free extract, 1% ether extract, 34% crude fiber, 11% ash, 1.1% Ca, and 0.32% P on DM basis (commercially analyzed). As a supplementation strategy, animals in the respective group grazed under the same conditions, but were also individually fed 2 kg/animal daily with a supplement in mobile feeders throughout the grazing experiment. The supplement offered was adjusted throughout the experiment until reaching 4.5 kg/animal daily. The supplement amount was determined to target 1.2 kg/d of body weight gain. The number of experimental units (animal) per treatment is depicted in Table 1. The supplement offered was composed of (DM basis) 40.9% poultry litter, 50% rice grinding, 6% molasses, 1.5% NaCl, 1.6% macro/micro mineral packet, and 330 mg/animal daily of monensin sodium (Rumensin®, Elanco Animal Health, Indianapolis, IN, USA). The supplement offered contained 17% of CP, 2.58 Mcal of ME/kg of supplement, 1.3% of Ca, and 0.97% of P. The supplement offered was maintained until the onset of the rainy season (approximately mid-May). The live weight (BW) of each individual animal was determined after a 12 h solid and liquid fasting period on day 0 and day 165 by using an electronic cattle scale (Fairbank® model FB2255; Fairbanks Scales Inc. Overland Park, KS, United States). Upon reaching a BW endpoint of 480 kg, cattle were successively sent in six slaughter lots to an inspected slaughterhouse located in the midwestern city of Barquisimeto, Lara State, approximately 500 km away from the ranch. Muscle score and frame size score were evaluated before shipping following Venezuela Decreto 181 [19]. The interim ADG (ADG 1) was calculated from day 0 up to the end of supplementation (165 d) and for the whole fattening phase (ADG 2) (i.e., d-0 to harvest). Carcass-adjusted final BW was calculated from HCW divided by the average dressing percent across treatments and adjusted by a 4% pencil shrink. Carcass-adjusted ADG 2 was calculated from carcass-adjusted final BW, initial BW, and days on feed.

### 2.2. Carcass Evaluation and Fabrication

Slaughter, dressing procedures, and postmortem inspection followed the Venezuelan (COVENIN) standards [20,21]. The carcass evaluation and fabrication procedures have been described in previous reports [22,23,24]. Briefly, linear measurements (thoracic depth, thigh width, leg perimeter, length of pelvic limb, and carcass length) and conformation profile score (1 = Very convex, 2 = Convex, 3 = Rectilinear, 4 = Concave, 5 = Very concave) were determined at the harvest floor [25]. After 48 hours postmortem at 4 °C, the chilled carcasses were evaluated for external fat finish (i.e., the amount of subcutaneous fat cover; 1 = Extremely abundant, 2 = Abundant, 3 = Medium, 4 = Slight, 5 = Scarce) and adipose maturity (i.e., fat color, where 1 = Ivory white, 2 = Creamy white, 3 = Light yellow, 4 = Intense yellow, 5 = Orange) following the guidelines of Decreto Presidencial No. 1896 [26]. Additionally, the chilled carcasses were graded according to the procedure stipulated by the Venezuelan regulation [26] and that of the USDA [27]. Accordingly, USDA yield grade was estimated with the ribeye area (REA), adjusted fat thickness at the 12th rib, and the percentage of kidney, pelvic, and heart fat (KPH) [27], whereas USDA quality grade was estimated with the marbling levels and the physiological (overall) maturity scores [27]. In turn, the overall maturity score was computed by balancing the skeletal (bone) and lean maturity scores [27]. Afterward, carcasses were reduced to subprimal cuts [28] with the removal of excess subcutaneous fat, when present, leaving a maximum fat thickness of 0.64 cm on any cut. The muscle cuts from the two sides were individually weighed to determine their percentage yield in relation to the whole cold carcass, as well as the combined percentage of subprimal cuts according to their market value in Venezuela [5,22,23]. Co-products (clean bone and trimmable fat) were also computed as proportions (%) of the cold carcass weight. The international equivalences of the commercial names given in Venezuela to each cut have been previously reported [5,23,24,28,29]. 

### 2.3. Cookery, Panel Sensory and Texture Tests

Evaluation of cooking loss, cooking time, sensory traits, and Warner–Bratzler shear force measurements (WBSF) of meat samples followed the guidelines of the American Meat Science Association (AMSA, 2016 [30]). These methods have been previously described in detail [5]. Briefly, at 2-day postmortem, 2.5 cm-thick LM (*longissimus lumborum*) steaks were cut alternately from the strip loin subprimal and accordingly assigned for sensorial evaluation and WBSF tests. The steaks were identified, vacuum packaged, immediately frozen, and stored (−30 °C) until further analyses. Each raw steak was weighed before cooking on an open electric grill. Upon reaching a temperature of 70 °C in the approximate geometric center of the steak, it was removed from the grill. Cooking time, cooking loss, and WBSF were recorded following AMSA guidelines [30]. Six to eight 25–45-year-old highly experienced panelists [31], from both sexes and different levels of instruction, tasted about 12 samples per day. Each panelist was given two or three samples for assigning ratings to juiciness, muscle fiber tenderness, overall tenderness, amount of connective tissue, and flavor intensity, according to a descriptive, structured scale consisting of eight points, where 1 = extremely dry, extremely tough, excessive connective tissue, and extremely bland, respectively; and 8 = extremely juicy, extremely tender, negligible amount of connective tissue, and extremely intense, respectively.

### 2.4. Statistical Analysis

All data were analyzed using the statistical program R Core Team [32]. The Shapiro–Wilk normality test [33] was performed for each response variable. Natural logarithm (y * = log (y) transformations were performed prior to the analysis when the variances were not homogeneous. The analysis of variance (ANOVA) was performed using the generalized linear model (GLM) with sex class, supplementation, and implant treatment as the main factors and their interactions for the growth performance variables, carcass traits, WBSF, and cookery traits. For individual and combined yield (%) of subprimal cuts, a linear mixed model (LMM) was applied to a completely randomized design with a 2 × 2 × 2 factorial arrangement. Fixed effects were supplementation treatment, implant, castration (sex class), and the first-order interactions, whereas the date of shipping was included as a random effect. For sensory traits, a similar LMM was used, including panelists as a random variable. Multiple mean comparisons were made by using the Tukey–Kramer test for unbalanced data [34] with a significance level of 0.05. For carcass grading performance, analysis of the frequency distribution of Venezuelan quality categories and USDA (quality and yield) grades were performed by sex class, implant, and supplementation treatment, and values were compared using the chi-square option of R Core Team [32] with a significance level of 0.05.

## 3. Results and Discussion

### 3.1. Growth Performance

No interactions (*p >* 0.05) between sex class × implant, sex class × supplementation, and supplementation × implant were observed for hip height, muscle score, frame-size score, BW, time to reach the endpoint, chronological age, adjusted final BW, and adjusted ADG2 (Appendix A). An interaction between sex class × implant was observed (*p* = 0.01) for the interim (non-adjusted) ADG1 (d0 to d163) and ADG2 (d0 to end), in which implanted steers had 27.5% and 28.2% greater ADG1 and ADG2 compared to their non-implanted counterparts, while implants did not affect bulls (Figure 1 and Figure 2). 

Authors cited by Lee et al. [35] pointed out additive anabolic effects under intensive feeding by implanting feeding steer calves with an EB–TBA combination. In addition, the EB–TBA implant has improved growth traits in grazing steers and generated a marginal income that justified its use [10]. Corroborating the current experiment, pioneering studies with bulls in South Carolina showed an absence of positive responses to EB–TBA for the rate of gain of such sex classes [35,36]. Very few studies in Venezuela [37,38] have been conducted to assess the response in growth rate to EB–TBA from grazing cattle. Current research findings agree with those of Araujo-Febres and Pietrosemoli [37], who reported a 30.4% improvement in the growth rate of Zebu crossbred steers under pasture supplementation with single implantation of EB–TBA. For non-supplemented cattle, a more attenuated significant response in growth rate to EB–TBA for grazing steers (18.6%) and a lack of positive responses from grazing bulls have been reported [38].

#### 3.1.1. The Main Effects of Castration

Hip height was the only live animal measurement significantly affected by castration. On average, steers were 1.8 cm taller (*p* = 0.03) than bulls at the hip (Table 2). Castrated animals being taller than their bull counterparts at comparative ages has been previously reported [39,40]. The suppressed secretion of androgens with castration leads to this prolonged growth of the epiphyseal plate that results in a disproportionate increase in the long bones in castrated males [39]. By design, animals were harvested when body weight reached 480 kg. However, the lack of statistical evidence to support effects or clear trends in favor of bulls in BW at the completion of the supplementation period or in days to reach the BW endpoint (Table 2) was unforeseen, because many studies (under intensive systems though) concur that bulls usually outperform steers in rate of gain [41,42,43]. There is a consensus that the advantageous performance of bulls is mainly due to the anabolic effects of testosterone but also estradiol produced in the testes [43]. Current results, however, are in line with those of Costa Rican [4] and Brazilian [7] workers, who did not find differences in final BW between bulls and steers grazing tropical pastures. Rodríguez et al. [4] summarized the possible causes of the paradoxical results between grazing and feedlot trials when bulls are compared to steers, as follows: (a) bulls require greater energy for maintenance, thus if nutritional needs are not met under grazing conditions bulls and steers grow at a similar rate; (b) the high temperature pattern in the tropics affects the ruminants’ utilization of low-quality forages by impairing the balance of nutrients needed for anabolic functions; and (c) greater metabolic action is needed to increase heat dissipation, thus requiring greater energy for maintenance [4].

#### 3.1.2. Main Effects of Supplementation

Cattle that were offered the supplement, which consisted of a source of ruminally degradable protein (74.5% of the CP), and highly fermentable carbohydrates (molasses) increased (*p* < 0.01) ADG1 by 24.9% compared to the non-supplemented group. Supplementation also increased (*p* < 0.01) the ADG2 by 8.63% and shortened (*p* < 0.01), by approximately 73 days, the length of time to reach the BW endpoint. At the end of the supplementation period, the cattle that were offered the supplement were 40.7 kg heavier (*p* = 0.02) than the non-supplemented group. On shipping to harvest, the supplemented group maintained this advantage in BW at a 2.4-mo younger age (*p* < 0.01) (Table 2). 

In tropical systems, the use of more affordable sources of NPN or protein can enhance the use of strategic supplementation to rangeland cattle consuming low-protein forages (<7% CP) during a prolonged dry season [11,44]. Numerous studies indicate the benefits of combining NPN and rumen-degradable true protein sources with highly fermentable carbohydrates on body weight gain as a consequence of greater forage consumption and the enhanced synthesis of ruminal microbial protein [45].During the final phase of finishing, the microbial crude protein may supply 90% of the metabolizable protein requirements to allow weight gain [46]. Similar results to the present experiment were reported by Tobias et al. [12] and Saddy et al. [47] in Zebu crossbred cattle (ADG of 0.80 and 0.68–0.80 kg, supplemented with 61.5 and 50% poultry litter, respectively). Variable ADG can be observed by beef cattle supplemented with poultry litter, while both the combination of ingredients inside supplements and the level of supplementation intake may affect grazing cattle growth performance [13,48,49,50,51].

### 3.2. Carcass Performance

No interactions (*p* > 0.05) between sex class × implant, sex class × supplementation, and supplementation × implant were observed for HCW, hot carcass dressing, cold carcass weight, conformation score, finish score, skeletal maturity, lean maturity, overall maturity, adipose maturity, ribeye area, 12th rib fat thickness, marbling score, thigh width, pelvic limb length, carcass length, and thoracic depth (Appendix A). An interaction between sex class × supplementation was observed for cold carcass dressing (*p =* 0.02) and KPH percentages. The cold carcass dressing yield did not vary with supplementation treatment in steers, while bulls offered the supplement dressed 2.63% greater than the non-supplemented bulls and 1.61% greater than non-supplemented steers (Figure 3). Supplemented steers showed greater percentage values of KPH than non-supplemented steers (*p* = 0.04) and supplemented bulls (+0.89%) (Figure 4). The current literature lacks assessments of carcass traits of grass-fed beef cattle with and without supplementation, although energy partitioning and additional deposition of adipose tissue as KPH for supplemented animals may be expected for beef cattle receiving sufficient metabolizable energy allowable for gain [18]. 

#### 3.2.1. Main Effects of Castration

Steer carcasses had a more (*p* = 0.02) desirable finish score (“Abundant” to “Medium”) and thicker backfat (*p* < 0.01) than those from bulls, while bulls showed more advanced (*p* = 0.02) skeletal and lean maturities and overall maturity (*p* < 0.01) (Table 3). In addition, steer carcasses exhibited longer (*p* = 0.03) pelvic limbs. The aforementioned observation aligns with their taller body stature at the hip, as shown in Table 2. No other main effects of sex class were observed on carcass characteristics (*p* ≥ 0.16). However, bulls were noted to tend (*p* = 0.08) to reach heavier HCW with a slightly better (lower numerical) conformation score (*p* = 0.11) and larger ribeye areas (*p* = 0.06). Except for backfat thickness, current research findings concur with Rodríguez et al. [4], who did not detect differences in HCW, dressing percentage, or ribeye area between grass-fed bulls and steers under tropical conditions. 

Likewise, Aricett et al. [7] reported similar HCW between bulls and steers fattened on pasture, but dressing percentage, backfat thickness, and marbling scores were greater for steers, whereas the LM area was larger for bulls [7]. In addition, it has been reported that bull carcasses showed larger LM areas and lower fatness levels than steers at the same endpoint [43,52,53]. Wang et al. [53] reported that the increased fat deposition in steers compared to bulls may be promoted in the liver by fatty acid binding protein-1, which was more expressed in steers than bulls. Moreover, in bulls, testosterone binds to receptors in muscles and stimulates increased incorporation of amino acids into protein, thereby increasing muscle mass without a concomitant increase in adipose tissue [54]. Current experiment findings indicate that steer carcasses were less mature than those of bulls, as indicated by the skeletal, lean, and overall maturity assessments (Table 3). Such findings agree with studies reviewed by Seideman et al. [41], indicating that bull carcasses are more physiologically mature than steers at the same age, according to the ossification patterns and lean color. At 12 months of age, differences between bulls and steers in cartilage development were negligible; however, at older ages, bull carcasses consistently exhibited a more advanced maturity [41]. Jacinto-Valderrama et al. [55], working with Nellore cattle, also reported that the bone physiological maturity of carcasses revealed that bulls, regardless of the production system (grazing vs. semi-confinement), were significantly older than immuno-castrated steers.

#### 3.2.2. Main Effects of Supplementation

Carcasses from supplemented cattle showed slightly greater levels (numerically lower scores) of marbling (*p* = 0.01), although the amounts of marbling were rather scarce in both supplemented and non-supplemented groups. Except for Jerez-Timaure and Huerta-Leidenz [15], no reports were found in the literature that assessed the effects of supplementation on carcass traits of grass-fed Brahman cross cattle. After providing a similar supplementation strategy used in the current experiment, Jerez-Timaure and Huerta-Leidenz [15] reported that supplement-offered bulls exhibited heavier carcass weight, greater dressing yield, younger overall maturity, and thicker backfat than their non-supplemented counterparts.

#### 3.2.3. Main Effects of Implant

No effects of implant treatment on carcass traits were detected (*p* ≥ 0.19). However, implanted cattle tended to have more advanced lean (*p* = 0.06) and overall (*p* = 0.07) maturities with negligible changes in marbling levels (*p* = 0.09). To our knowledge, there are no previous studies assessing the effects of a single EB–TBA implant on the carcass characteristics of Brahman cross cattle fattened under tropical conditions. Under similar animal and environmental conditions to the current experiment, two aggressive implant regimens (consisting of (1) double dose of zeranol (72 mg) with reimplantation at 90 d, and (2) mixed treatment consisting of EB–TBA during the first 90d and reimplantation with zeranol at double dose on day 90) did not differ in their effects on carcass characteristics [56].

No differences (*p >* 0.05) in the frequencies of USDA grades/Venezuelan categories were detected between sex classes or treatments (data not shown), but some trends deserve discussion. None of the steer carcasses reached the top-quality Venezuelan grade (AA), while four bull carcasses were categorized as C (the fourth Venezuelan quality grade). Eighteen out of 24 steers were USDA Standard and six were graded USDA Select. Given that 14 bulls had a chronological age of fewer than 30 months, their carcasses fell into the first (youngest) maturity classification (A), hence they were eligible for the “Bullock” class designation [27]. Only one non-implanted, supplemented bull was graded Select, whereas 14 were graded Standard. Because eight bulls displayed characteristics of more advanced (>30 mo of age) carcass maturities, they would be designated as “Bulls” with no eligibility to be quality-graded in the USA [27]. On the other hand, regardless of supplementation or implant treatment, USDA yield grade (YG) 2 prevailed in the steer and bull samples. Other experiments in the American tropics or subtropics using grazing young bulls with different degrees of Zebu genetics [15,23,24,57] reported that these cattle performed poorly in carcass quality (i.e., they did not exceed the "Slight" marbling level, and most were graded as USDA Standard). Conversely, the USDA yield grade performance of bull carcasses under tropical conditions has been remarkably superior under different feeding conditions [15,22,24,58] by reaching the top two USDA yield grades (YG 1 or YG 2).

### 3.3. Effects on Yields of Beef Subprimals and Coproducts

No interactions (*p ≥* 0.07) between sex class × supplementation and supplementation × implant were observed for individual and combined yields (%) of subprimal cuts (Appendix A). An interaction between sex class × implant was observed for bone-in brisket (*p* = 0.04) and bone-in hind shank (*p* = 0.04), in which the implant treatment decreased the yield (%) of bone-in brisket in the group of steers (Figure 5) and increased the yield (%) of bone-in hind shank in the group of bulls (Figure 6). In turn, the non-implanted group of bulls yielded a lower (*p* < 0.05) percentage of bone-in brisket (Figure 5) and bone-in hind shank (Figure 6) than steers.

#### 3.3.1. Main Effects of Sex Class

Steer carcasses yielded slightly higher (<0.5%) proportions of tenderloin, knuckle, and top round and 1.64% more higher-value boneless cuts than those from bulls (*p* < 0.01). Conversely, bull carcasses outperformed those of steers in proportions of chuck roll (+3.34%), medium-value boneless cuts (+3.47%), and the yield of total cuts (+1.35%) with a lesser proportion (−1.23%) of trimmable fat (*p* < 0.01) (Table 4). The proportions of high-value, boneless lean cuts and total cuts were 1.64% and 1.35% higher in steers, respectively, whereas bull carcasses yielded more (+ 3.47%) medium-value cuts and lower proportions of trimmable fat (−1.24%) and clean bone (−0.31%) (Table 5). 

#### 3.3.2. Main Effects of Implant and Supplementation

Contrary to expectations, carcasses from implanted cattle did not (*p* ≥ 0.13) differ in yield percentage of most subprimal cuts with respect to the non-implanted group. In fact, the non-implanted group had slighter greater (*p* < 0.01) yields of low-value cuts than the implanted group. With the only exception of boneless top round that showed a small yield reduction (−0.29%; *p* < 0.01) with cattle offered the supplement, no (*p* ≥ 0.13) individual subprimal cuts were affected by the supplementation treatment (Table 5). The non-significant reductions in the carcass yield of high-value boneless lean cuts (−0.69%, *p* = 0.70 ), medium-value cuts (−1.32%, *p* = 0.37), and total cuts (−1.52%, *p* = 0.42) in the supplemented group could be due to its greater (*p* < 0.01) proportion (+1.40%) of trimmable fat and a trend to exhibit a higher proportion of low-value cuts (+0.49%, *p* = 0.10) (Table 5). These findings agree with those reported by Jerez-Timaure and Huerta-Leidenz [15] for carcasses of grazing bulls offered a similar supplement in the same ranch. Likewise, in the latter study [15], the bulls offered such a supplement yielded more trimmable fat and had a lower carcass yield of high-value, medium-value, and total cuts compared to a non-supplemented group. Despite the fact that the carcass cut-out yield should strongly attract interest from the meat industry, compositional commercial traits are seldom reported by Latin American workers. Therefore, the aforementioned effects of treatments on fabrication variables could not be further and appropriately discussed. With grain-fed cattle, Foutz et al. [59] compared different types of implants on the percentage yield of selected subprimals and reported that, in general, all implants significantly increased subprimal yields, with the largest improvements by combining an estrogen with TBA (i.e., EB–TBA). The current experiment findings observed for subprimal cuts are not in agreement with those reported by Foutz et al. [59]. 

### 3.4. Effects on Cookery and Meat Quality Traits

No interactions (*p >* 0.05) between sex class × implant, sex class × supplementation, and supplementation × implant were observed for juiciness, connective tissue, tenderness, flavor intensity, WBSF, cooking loss, or cooking time (Appendix A). 

#### 3.4.1. Main Effects of Sex Class

Unaged (2-d postmortem) LM steaks from steers lost 4.44% more (*p* < 0.01) weight upon cooking and panelists rated them higher (*p* < 0.01) for muscle fiber tenderness, overall tenderness (*p* < 0.01), and amount of connective tissue (*p* < 0.01) compared to bulls (Table 6). Although statistically significant, the small magnitude (<0.5 units) of these sensory differences and the lack of difference in WBSF values denote samples of similar textural characteristics. Furthermore, this sample of Brahman-influenced, grass-fed cattle would not reach the average tenderness threshold required to please consumers in Venezuela (WBSF < 4.09 kg; [60]). The superiority in sensory attributes of steers over bulls has been reported in other studies [61,62], and differences due to castration have been more notorious when cattle were intensively fed [4]. Rodriguez et al. [4] studied the effect of castration on the growth performance of Brahman-crossed cattle fattened on grass in Costa Rica, where panelists’ ratings tended to favor 14-d. aged LM steaks from early (3 mo-castrated) steers, but their sensory differences against bulls did not differ (*p* < 0.05). Additionally, in Brazil, it has been reported that mean panelist ratings for tenderness, juiciness, and flavor, or cooking losses, differed significantly between bulls and steers [63]. Nevertheless, Rodriguez et al. [4] reported that LM steaks from bulls required significantly greater WBSF than those from all groups of steers castrated at 3, 7, or 12 months of age, which is consistent with the observations of other studies cited by the same authors for cattle under grazing in tropical conditions in Costa Rica. According to Silva et al. [61], the lack of differences between bulls and steers in WBSF and other meat quality variables of unaged meat (1-d postmortem) became significant at 14 d postmortem. These workers [61] reported that 14-d aged meat from Nellore steers exhibited significantly lower WBSF, greater myofibrillar fragmentation index, and greater postmortem desmin degradation compared to those from bulls. 

Cattle offered the supplement did not exhibit expected beneficial effects on beef cookery or palatability traits. Small yet statistically significant differences (*p* ≤ 0.01) in panelist ratings for the amount of connective tissue (0.52 units), muscle fiber tenderness (0.30 units), and overall tenderness (0.53 units) were in favor of samples from the non-supplemented group, which is in accordance with a previous study with Brahman-influenced cattle pasturing in the same ranch [15]. In the latter experiment [15] using a similar supplement, the authors reported that the supplementation significantly increased WBSF in 0.91 kg and decreased panelist ratings (in approximately 0.45 units) for muscle fiber tenderness, overall tenderness, and the amount of connective tissue. However, the flavor ratings were not significantly affected by the supplementation treatment [15]. In this respect, Jeremiah and Gibson [14] evaluated the flavor profile and sensory traits of beef from poultry litter-based supplemented cattle and detected little difference with the control group in the incidence of an inappropriate off-flavor and the amount of connective tissue. The differences observed by these authors [15] were of insufficient magnitude to impact the texture or flavor amplitude. Either way, there was a lack of difference in consumer acceptability and sensory traits evaluated by the trained descriptive panel. 

#### 3.4.2. Main Effects of Implant

The LM steaks from the implanted cattle did not (*p* ≥ 0.45) affect cookery, sensory traits, or WBSF compared to the non-implanted group. The current experiment observations in regard to WBSF and sensory traits agree with the metanalytical study performed by Lean et al. [8]. According to these authors, tenderness, flavor, juiciness, and connective tissue amount in beef are not associated with the use of hormonal implants, and that the use of a single implant, whether this be a single androgenic agent (e.g., TBA) or in combination with an estrogenic compound (i.e., EB–TBA), had a limited effect on WBSF [8].

## 4. Conclusions

Inadequate management and feeding practices of breeders operating in neotropical Apure’ savannas impedes adoption of a cow-calf-to-finish system and attainment of the most profitable markets with pasture-fed cattle of competitive traits in beef yield and quality. The castration of calves at 7 months of age could be performed without detrimental consequences to cattle growth performance while eliciting a growth enhancement to a single dose implant with EB + TBA, improving carcass finish, the yield of high-value subprimals, and meat tenderness. Supplementation improved key performance traits in cattle and decreased age at slaughter, which is clearly advantageous for rotating more animals per production cycle. Except for the increased dressing percentage in bulls, supplementation marginally influenced carcass traits, while it negatively affected tenderness-related sensory traits. The implant enhanced the rate of gain of steers only, without inducing adverse effects on palatability or beneficial effects on cut-out yields. The main limitation of the present study is the relatively low number of experimental units per treatment. Further studies with larger sample sizes are needed to reevaluate these effects with a prolonged time on pasture supplementation to reach heavier BW and/or thicker backfat thickness.

## Figures and Tables

**Figure 1 animals-11-03441-f001:**
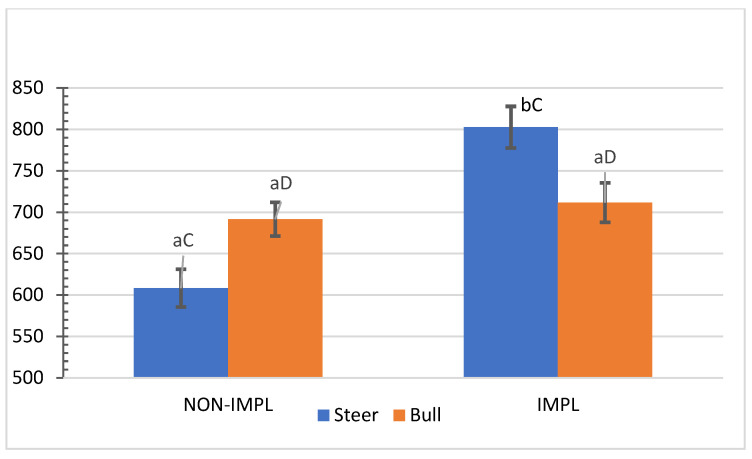
Mean values ± standard error for sex class × implant interaction (*p* = 0.01) for average daily gain from day 0 to day 163 (ADG1). IMPL: Implantation of a hormonal compound containing 24 mg estradiol benzoate 17 ß and 120 mg of trenbolone acetate. NON-IMPL: No implant. Bars with a common superscript lowercase letter (a, b) for implant treatments within the same sex class do not differ (*p* > 0.05) and bars with a common superscript uppercase letter (C, D) for sex classes within the same implant treatment, do not differ (*p* > 0.05).

**Figure 2 animals-11-03441-f002:**
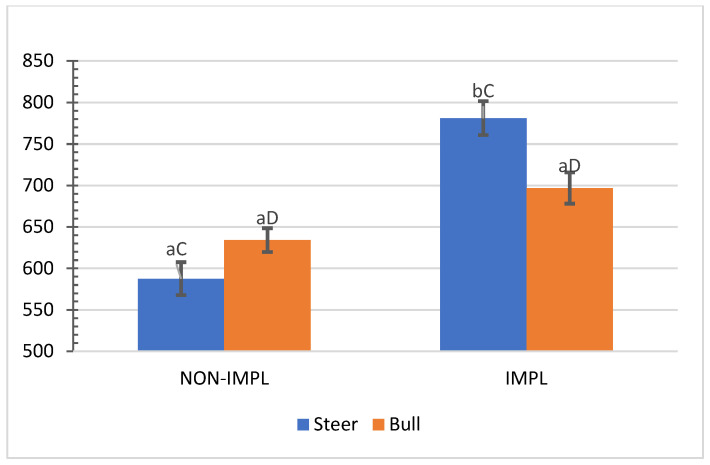
Mean values ± standard error for sex class × implant interaction ((*p* = 0.01) for average daily gain from day 0 to shipment day (ADG2). IMPL: Implantation of a hormonal compound containing 24 mg estradiol benzoate 17 ß and 120 mg of trenbolone acetate. NON-IMPL: No implant. Bars with a common superscript lowercase letter (a, b) for implant treatments within the same sex class do not differ (*p* > 0.05) and bars with a common superscript uppercase letter (C, D) for sex classes within the same implant treatment, do not differ (*p* > 0.05).

**Figure 3 animals-11-03441-f003:**
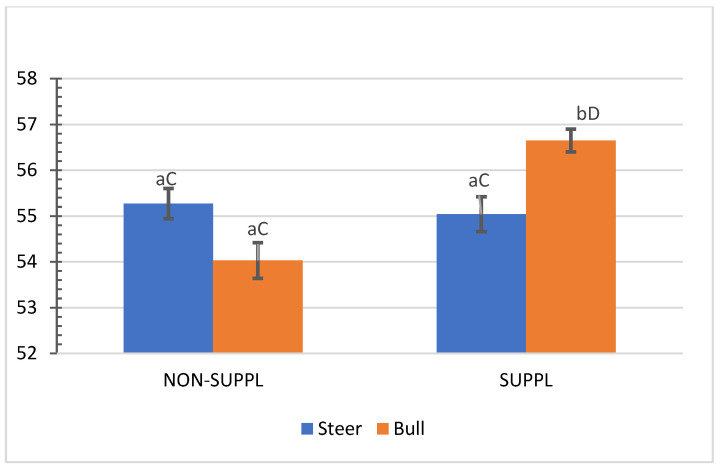
Mean values ± standard error for sex class × supplementation interaction (*p* = 0.02) for cold carcass dressing (%). SUPPL: Pasture supplementation with a poultry litter-based (PLB) supplement. NON-SUPPL: No PLB supplement, only minerals. Bars with a common superscript lowercase letter (a, b) for supplementation treatments within the same sex class do not differ (*p* > 0.05) and bars with a common superscript uppercase letter (C, D) for sex classes within the same supplementation treatment, do not differ (*p* > 0.05).

**Figure 4 animals-11-03441-f004:**
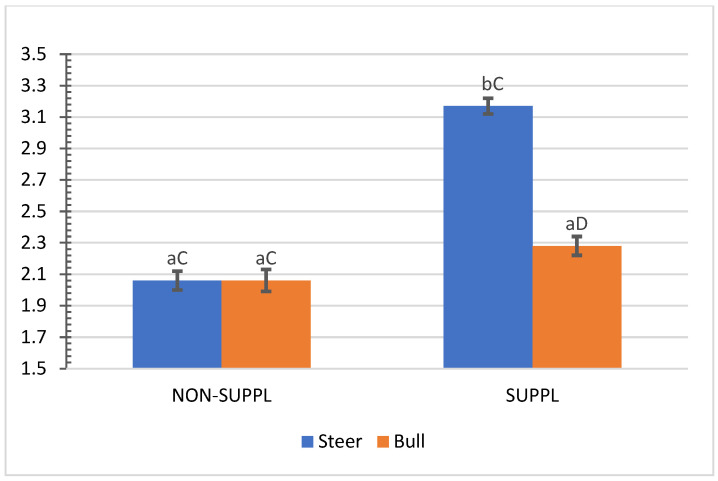
Mean values ± standard error for sex class × supplementation interaction (*p* = 0.04) for kidney, pelvic and heart fat (KPH) percentage. SUPPL: Pasture supplementation with a poultry litter-based (PLB) supplement. NON-SUPPL: No PLB supplement, only minerals. Bars with a common superscript lowercase letter (a, b) for supplementation treatments within the same sex class do not differ (*p* > 0.05) and bars with a common superscript uppercase letter (C, D) for sex classes within the same supplementation treatment, do not differ (*p* > 0.05).

**Figure 5 animals-11-03441-f005:**
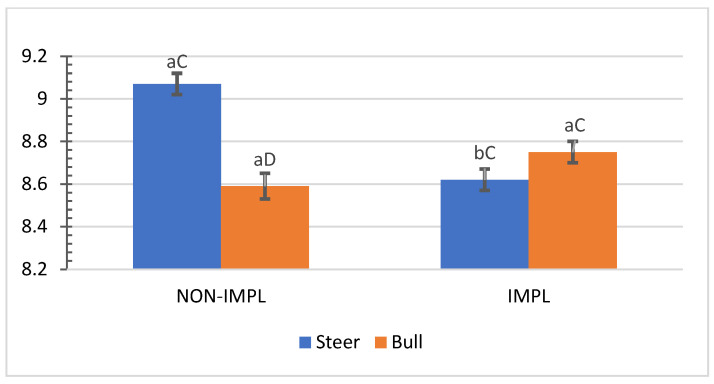
Mean values ± standard error for sex class × implant interaction (*p* = 0.04) for yield (%) of carcasses in bone-in brisket. IMPL: Implantation of a hormonal compound containing 24 mg estradiol benzoate 17 ß and 120 mg of trenbolone acetate. NON-IMPL: No implant. Bars with a common superscript lowercase letter (a, b) for implant treatments within the same sex class do not differ (*p* > 0.05) and bars with a common superscript uppercase letter (C, D) for sex classes within the same implant treatment, do not differ (*p* > 0.05).

**Figure 6 animals-11-03441-f006:**
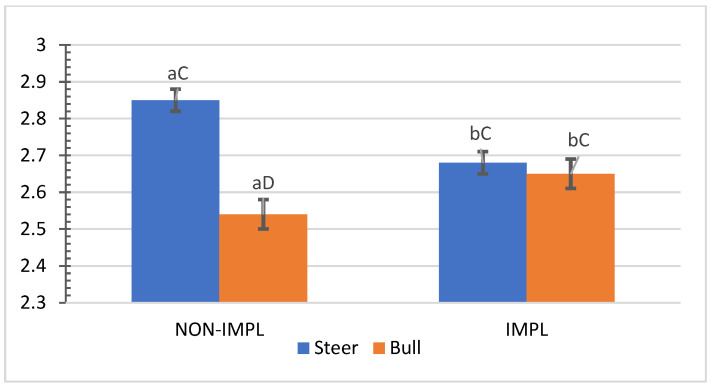
Mean values ± standard error for sex class x implant interaction (*p* = 0.04) for yield (%) of carcasses in bone-in hind shank (BIHH). IMPL: Implantation of a hormonal compound containing 24 mg estradiol benzoate 17 ß and 120 mg of trenbolone acetate. NON-IMPL: No implant. Bars with a common superscript lowercase letter (a, b) for implant treatments within the same sex class do not differ (*p* > 0.05) and bars with a common superscript uppercase letter (C, D) for sex classes within the same implant treatment, do not differ (*p* > 0.05).

**Table 1 animals-11-03441-t001:** Number of experimental units within each treatment factor ^a^.

Sex Class	Implanted ^b^		Non-Implanted	
	Supplemented ^c^	Non-Supplemented	Supplemented ^c^	Non-Supplemented
Bull	5	6	6	6
Steer	6	6	6	6

^a^ Figures indicate number of animals in each treatment group, ^b^ Group implanted with a hormonal implant containing 24 mg estradiol benzoate 17 ß (EB) and 120 mg of trenbolone acetate (TBA), ^c^ Supplemented group received pasture supplementation with a poultry litter-based supplement. Total number of experimental units (*n* = 47).

**Table 2 animals-11-03441-t002:** Main effects of sex class, supplementation, and implant treatments on growth performance of grass-fed Brahman males.

Variables	Sex Class		Supplementation (SUPPL)		Implant		SEM	*p*-Value		
	Steer (*n* = 24)	Bull (*n* = 23)	NON-SUPPL ^a^ (*n* = 24)	SUPPL ^b^ (*n* = 23)	NON-IMPL ^c^ (*n* = 24)	IMPL ^d^ (*n* = 23)		Sex Class	SUPPL	Implant
Hip height, cm	137.7	135.9	136.7	136.9	136.2	137.4	0.48	0.03	0.55	0.28
Muscle score ^e^	3.79	3.73	3.79	3.73	3.75	3.78	0.06	0.37	0.38	0.37
Frame-size score ^f^	2.58	2.91	2.75	2.74	2.66	2.82	0.07	0.21	0.54	0.09
Chronological age, mo	30.90	30.43	31.85	29.44	30.75	30.59	0.23	0.79	<0.01	0.20
BW at the end of SUPPL period, kg.	446.0	461.4	433.0	473.7	448.2	458.5	4.06	0.74	0.02	0.26
Final BW at shipping d ^g^, kg	470.3	483.1	464.1	479.1	469.1	484.3	4.08	0.32	0.34	0.31
Time to reach endpoint, d	210.4	211.4	246.75	173.5	212.3	209.4	6.68	0.39	<0.01	0.21
ADG1 (d 0—d 163), g	705.5	701.6	616.2	791.1	648.2	759.1	24.37	0.12	<0.01	<0.01
ADG2 (d 0—d of shipping), g	684.4	664.1	574.0	779.2	610.8	740.9	25.60	0.27	<0.01	<0.01
Adjusted BW, kg ^h^	451.4	463.8	455.2	459.9	450.3	464.9	4.34	0.32	0.34	0.31
Adjusted ADG2, g ^i^	590.7	568.4	495.3	667.9	518.4	643.9	23.47	0.26	<0.01	0.03

^a^ Non-supplemented group, ^b^ Supplemented group received pasture supplementation with a poultry litter-based supplement, ^c^ Non-implanted group, ^d^ A hormonal implant containing 24 mg estradiol benzoate 17 ß (EB) and 120 mg of trenbolone acetate (TBA)., ^e^ 1 = very heavily muscled and 5 = lightly muscled [19], ^f^ 1 = very large and 5 = very small [19], ^g^ Shipping day was the date of loading cattle from the ranch to the abattoir after reaching the endpoint, ^h^ Carcass-adjusted final BW was calculated from HCW divided by the average dressing percent across treatments and adjusted by a 4% shrink, ^i^ Carcass-adjusted ADG2 was calculated from carcass-adjusted final BW, initial BW, and days on feed.

**Table 3 animals-11-03441-t003:** Main effects of sex class, supplementation, and implant treatments on carcass characteristics of grass-fed Brahman males.

Variables	Sex Class		Supplementation (SUPPL)		Implant		SEM	*p*-Value		
	Steer (*n* = 24)	Bull (*n* = 23)	NON-SUPPL ^a^ (*n* = 24)	SUPPL ^b^ (*n* = 23)	NON-IMPL ^c^ (*n* = 24)	IMPL ^d^ (*n* = 23)		Sex Class	SUPPL	Implant
Hot carcass weight, kg	266.1	274.0	265.6	274.6	266.1	274.1	2.43	0.08	0.53	0.19
Hot carcass dressing, %	56.60	56.67	56.06	57.33	56.71	56.66	0.40	0.68	0.56	0.61
Cold carcass weight, kg	259.3	266.8	258.8	267.2	259.3	266.8	2.38	0.84	0.07	0.62
Cold carcass dressing, %	55.15	55.28	54.65	55.80	55.27	55.16	0.86	0.27	0.74	0.51
Conformation score ^e^	3.08	3.22	3.37	2.91	3.08	3.21	0.07	0.11	0.22	0.84
Finish score ^f^	2.95	3.39	3.17	3.18	3.12	3.21	0.09	0.02	0.65	0.35
KPH, %	2.62	2.17	2.07	2.74	2.44	2.35	0.16	0.53	<0.01	0.56
Skeletal maturity ^g^	189.5	198.7	206.7	180.7	192.5	195.7	2.97	0.02	<0.01	0.25
Lean maturity ^g^	172.1	184.8	187.5	168.7	172.9	183.9	2.92	0.02	0.03	0.06
Overall maturity ^g^	181.5	191.5	189.3	183.5	183.5	189.3	2.58	<0.01	0.35	0.07
Adipose maturity ^h^	2.70	2.61	2.58	2.64	2.67	2.65	0.04	0.68	0.36	0.68
Ribeye area, cm^2^	67.80	71.81	69.11	70.43	68.52	71.05	0.52	0.06	0.85	0.25
12th rib fat thickness, mm	2.58	1.65	1.87	2.39	2.20	2.04	1.04	<0.01	0.16	0.28
Marbling score ^i^	4.66	4.95	4.95	4.65	4.70	4.91	0.17	0.87	0.01	0.09
Thigh width, cm	58.83	57.43	58.95	57.30	57.29	59.04	0.55	0.64	0.91	0.81
Pelvic limb length, cm	72.63	70.36	71.67	71.37	71.59	71.45	0.68	0.03	0.74	0.23
Carcass length, cm	128.1	130.2	129.2	129.1	128.6	129.7	0.33	0.31	0.77	0.78
Leg perimeter, cm	114.3	113.7	114.2	113.7	113.7	114.3	0.28	0.52	0.26	0.32
Thoracic depth, cm	36.70	36.26	35.12	37.91	36.50	36.47	0.37	0.16	<0.01	0.88

^a^ Non-supplemented group, ^b^ Supplemented group received pasture supplementation with a poultry litter-based supplement, ^c^ Non-implanted group, ^d^ A hormonal implant containing 24 mg estradiol benzoate 17 ß (EB) and 120 mg of trenbolone acetate (TBA), ^e^ Conformation scores: 1 = Very convex, 2 = Convex, 3 = Rectilinear, 4 = Concave, 5 = Very concave [25], ^f^ Finish score: 1 = Extremely abundant, 2 = Abundant, 3 = Medium 4 = Slight, 5 = Scarce [26], ^g^ Skeletal, lean, and overall maturity: 100–199: represent immature animals (100 is equal to A00 and 199 is equal to A99); 200–299: represent more mature animals (200 is equal to B00 and 299 is equal to B99) [27]. KPH: Kidney, pelvic and heart fat in percentage, ^h^ Adipose maturity: 1 = ivory white, 2 = creamy white, 3 = light yellow, 4 = intense yellow, 5 = orange, ^i^ Marbling scores: 1 = abundant to moderate, 2 = small, 3 = slight, 4 = traces, 5 = practically devoid [26].

**Table 4 animals-11-03441-t004:** Main effects of sex class, supplementation, and implant treatments on individual yield (%) of subprimal cuts of grass-fed Brahman males.

Variables ^a^	Sex Class		Supplementation (SUPPL)		Implant		SEM	*p*-Value		
	Steer (*n* = 24)	Bull (*n* = 23)	NON-SUPPL ^b^ (*n* = 24)	SUPPL ^c^ (*n* = 23)	NON-IMPL ^d^ (*n* = 24)	IMPL ^e^ (*n* = 23)		Sex Class	SUPPL	Implant
Tenderloin	2.19	2.09	2.19	2.10	2.16	2.12	0.04	0.02	0.47	0.52
Rib-eye roll and Strip-loin	8.49	8.19	8.26	8.43	1.35	1.84	0.28	0.48	0.46	0.25
Knuckle	3.83	3.58	3.80	3.61	3.77	3.64	0.10	<0.01	0.20	0.53
Center cut sirloin	3.11	2.96	3.09	2.97	3.04	3.01	0.08	0.33	0.83	0.11
Bottom (outside) round	3.65	3.36	3.54	3.47	3.48	3.53	0.13	0.16	0.29	0.97
Eye of round	1.85	1.84	1.86	1.84	1.83	1.87	0.07	0.96	0.75	0.67
Top sirloin cap or rump	1.73	1.64	1.68	1.67	1.66	1.69	0.09	0.37	0.16	0.58
Top (inside) round	6.87	6.45	6.81	6.52	6.65	6.68	0.14	<0.01	0.15	0.56
Shoulder clod with top blade	8.32	8.48	8.59	8.19	8.32	8.47	0.24	0.16	0.25	0.14
Chuck (mock) tender	1.03	1.05	1.07	1.01	1.00	1.01	0.04	0.35	0.55	0.92
Tri-tip	0.96	0.92	0.98	0.90	0.93	0.95	0.04	0.08	0.36	0.13
Chuck roll	12.12	15.46	14.16	13.33	13.45	14.07	0.63	<0.01	0.48	0.47
Heel of round	1.43	1.39	1.42	1.39	1.39	1.43	0.05	0.09	0.17	<0.01
Inside skirt, flank, rose meat	2.95	2.88	2.85	2.95	2.91	2.93	0.18	0.31	0.33	0.85
Rib plate	8.85	8.67	8.69	8.85	8.84	8.69	0.25	0.10	0.18	0.20
Bone-in brisket	5.95	5.93	5.85	6.03	6.13	5.74	0.24	0.24	0.12	0.13
Bone-in fore shank	1.78	1.73	1.70	1.81	1.75	1.76	0.08	0.46	0.95	0.48
Bone-in hind shank	2.76	2.59	2.70	2.66	2.70	2.66	0.12	0.79	0.31	0.42

^a^ Equivalence of names for individual cuts in different countries was reported by Montero et al. [28] and yield values are expressed as percentages of the cold carcass weight, ^b^ Non-supplemented group, ^c^ Supplemented group received pasture supplementation with a poultry litter-based supplement, ^d^ Non-implanted group, ^e^ A hormonal implant containing 24 mg estradiol benzoate 17 ß (EB) and 120 mg of trenbolone acetate (TBA).

**Table 5 animals-11-03441-t005:** Main effects of sex class, supplementation, and implant treatments on combined percentage of subprimal/retail cuts according to their market value in Venezuela for grass-fed Brahman males.

Variables ^a^	Sex Class		Supplementation (SUPPL)		Implant		SEM	*p*-Value		
	Steer (*n* = 24)	Bull (*n* = 23)	NON-SUPPL ^b^ (*n* = 24)	SUPPL ^c^ (*n* = 23)	NON-IMPL ^d^ (*n* = 24)	IMPL ^e^ (*n* = 23)		Sex Class	SUPPL	Implant
High-value boneless cuts ^f^	32.69	31.05	32.22	31.53	31.90	31.87	0.19	<0.01	0.70	0.52
Medium-value boneless cuts ^g^	22.91	26.38	25.25	23.93	24.22	25.01	0.34	<0.01	0.37	0.18
Low-value cuts ^h^	22.31	21.82	21.84	22.31	22.33	21.79	0.12	0.58	0.10	0.05
Total cuts ^i^	77.91	79.26	79.32	77.78	78.46	78.68	0.27	0.04	0.42	0.79
Trimmable fat	8.24	7.01	6.95	8.04	7.67	7.60	0.23	<0.01	<0.01	0.38
Clean bone	13.31	13.00	13.27	13.01	13.34	12.98	0.13	0.08	0.14	0.63

^a^ Values of composite groups of commercial cuts, trimmed fat, and clean bone are expressed as percentages of the cold carcass weight, ^b^ Non-supplemented group, ^c^ Supplemented group received pasture supplementation with a poultry litter-based supplement, ^d^ Non-implanted group, ^e^ A hormonal implant containing 24 mg estradiol benzoate 17 ß (EB) and 120 mg of trenbolone acetate (TBA), ^f^ High-value boneless cuts: tenderloin + rib-eye roll and strip-loin + center cut sirloin or top sirloin butt + eye of round + top (inside) round + bottom (outside) round + knuckle + tri-tip + heel of round, ^g^ Medium-value boneless cuts: shoulder clod and flat iron + chuck (mock) tender + chuck roll, ^h^ Low-value cuts: brisket + inside skirt, flank, flank steak, rose meat and shoulder rose + rib plate + fore shank + hind shank, ^i^ Total cuts: consists of the sum of the high-, medium-, and low-valued cuts.

**Table 6 animals-11-03441-t006:** Main effects of sex class, supplementation, and implant treatments on meat quality traits of grass-fed Brahman males.

Variable	Sex Class		Supplementation (SUPPL)		Implant		SEM	*p*-Value		
	Steer (*n* = 24)	Bull (*n* = 23)	NON-SUPPL ^a^ (*n* = 24)	SUPPL ^b^ (*n* = 23)	NON-IMPL ^c^ (*n* = 24)	IMPL ^d^ (*n* = 23)		Sex Class	SUPPL	Implant
Juiciness ^e^	4.50	4.61	4.46	4.63	4.58	4.52	0.15	0.18	0.08	0.48
Amount of connective tissue ^f^	3.89	3.50	3.97	3.45	3.69	3.70	0.22	<0.01	<0.01	0.85
Muscle fiber tenderness ^g^	4.29	3.97	4.29	3.99	4.10	4.16	0.21	<0.01	0.01	0.67
Overall tenderness ^g^	4.04	3.62	4.11	3.58	3.84	3.83	0.24	<0.01	<0.01	0.81
Flavor intensity ^h^	5.81	5.85	5.85	5.81	5.82	5.84	0.07	0.44	0.62	0.77
WBSF ^i^, kg	5.03	5.26	4.91	5.38	4.92	5.37	0.05	0.78	0.15	0.97
Cooking loss, %	41.80	37.36	39.96	39.28	39.86	39.38	0.99	<0.01	0.82	0.70
Cooking time, min.	100.5	100.5	101.8	99.2	98.8	102.4	2.82	0.35	0.43	0.97

^a^ Values of composite groups of commercial cuts, trimmed fat, and clean bone are expressed as percentages of the cold carcass weight, ^b^ Non-supplemented group, ^c^ Supplemented group received pasture supplementation with a poultry litter-based supplement, ^d^ Non-implanted group, ^e^ 8-point hedonic scale, where 1 = extremely dry and 8 = extremely juicy, ^f^ 8-point hedonic scale, where 1 = abundant amount of connective tissue and 8 = no connective tissue, ^g^ 8-point hedonic scale, where 1= extremely tough and 8 = extremely tender, ^h^ 8-point hedonic scale, where 1= extremely bland and 8 = extremely intense, ^i^ Warner–Bratzler shear force.3.4.2. Main Effects of Supplementation.

## Data Availability

Data are not available in public datasets, please contact the authors.

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
