# Peer review of "Effects of Sex Class, a Combined Androgen and Estrogen Implant, and Pasture Supplementation on Growth and Carcass Performance and Meat Quality of Zebu-Type Grass-Fed Cattle"

_animals, 2021, doi:10.3390/ani11123441_

Round 1

Reviewer 1 Report

The results of the research is interesting and useful to beef cattle production in the tropical ecosystems throughout the world. The content of the manuscript fits the journal topics well. However, the manuscript was not well written. Some points which need to be revised before it can be accepted for publication.

  1. Lines 95-147: Materials and Methods were not clearly described. I would suggest re-write this part and describe the animals, pasture, concentrates and experimental design with subtitles. The components of the supplemented concentrate should be described.
  2. The methods and references for analyzing the nutritional composition including dry matter, crude protein, organic matter, neutral detergent fiber and acid detergent fiber and energy of the pasture and the concentrates should be described in detail.

3.Statistical analysis for the interactions among gender, supplementation and implant should be carried out and include the results in the Results and Discussion section.

  1. In the Results and Discussion section: The intakes of dry matter intake and the major nutrients such as crude protein, energy, organic matter, etc, should be added.
  2. Any difference between "Steer" and "Bull"? Steer and bull have the same meaning. Please check the Tables and Figures.
  3. The English of the manuscript needs to be improved.
  4. In the Abstract and Conclusions sections, I would suggest draw a clearer conclusion, and indicate how to balance these three factors in grass-fed cattle production.
  5. Other minor points:

L 40: Define the abbreviation "ADG".

L42: Remove the space.

L67-69: Change to‘……non-implanted counterparts[4,5,6]’

L71-76: This is a long sentence. Suggest divide it into two sentences.

L80: “has shown to be beneficial”.

L82-84: Add a reference.

L89: Change to “Materials and Methods”.

L98: The size of animal experiment was 47, why not using 48 to keep the same size in each treatment.

L130: How were the bulls or steers weighed?

L323: Change to "kg".

L422: Change to "Main effects of supplementation".

L549: New line after the subtitle.

L614: Delete the reference[15].

Reviewer 2 Report

The aim of the research was to evaluate the effects of sex class, a combined androgen and estrogen implant, and pasture supplementation on growth and carcass performance, and meat quality of Zebu-type grass-fed cattle. The number animals used in the experiment is sufficient. The applied research methods are correct. The discussion is well conducted and comprehensive. Well-chosen references. Before publishing in Animals, the article requires additions and corrections. The proposed changes are listed below:

General comments:

Please prepare the article in accordance with the instructions for authors.

For affiliates, the first name and surname initials for each co-author of the article should be provided, the same as given in the "Author Contributions" chapter.

In tables 1-6, the table header must be in bold

The pages of the article must be correctly numbered from 1 of 28 to 28 of 28

The Animals and mdpi logo must be removed from pages 9, 10, 12, 14, 18, 20, etc.

The "Author Contributions" chapter must follow the instructions: activity 1 name, authors, activity 2, authors, etc.

In the References section, for a range of pages, use the long "-" from the insert function for all References items

Detailed comments:

L23-24, please remove the sentence "This study ...." The purpose of the work was given twice. One is redundant

L28 carcass finish? But what about the finish?

L43 BW….. must be in  line 42

L45, 46 too many spaces

L52 without a „dot” after "class"

L60 and others [1,2], no spaces between References numbers

L76 too many spaces before "Because .."

L89 „Materials and Methods” instead of current form

L102 without ")" after [18]

L112 „%” with the number of

L132-133 sum of components should equal 100%

L135 „2.97 Mcal / kg of ME” is correct?

L162 [5,23-24,28-29] ??

L 203 p > 0.05 instead of current form

L204-205 "chronological age" also

L265 [41-43] instead of current form

L284 40.7 kg or 40.6 kg?

L330 (p > 0.05) instead of current form

L389 thoracic depth p = 0.16, but overall maturity p < 0.01

L393 for conformation score (p = 0.11) or p = 0.10?

L423 where there is a description for lean maturity, skeletal maturity, and KPH?

L 509 provide the number of the Tables

L516 1.23% or 1.24%?

L521 low-value cuts is in Table 5 not Table 4

L525 significant ??, see Table 5, description not consistent with results

L549 add one line before L549 without text

L816 „Plos One” instead of current form

L839 „Livest. Sci.” instead of current form

Reviewer 3 Report

The paper “Effects of sex class, a combined androgen and estrogen im- plant, and pasture supplementation on growth and carcass performance, and meat quality of Zebu-type grass-fed cattle” by Huerta-Leidenz et al evaluates some practices (castration, estrogen implants and pasture integration) on cow-calf finish system. Although no very new information are reported in the paper, it is interesting and well structured. Two are the main concerns of this reviewer on the paper. Firstly, the number of animals for each treatment is limited: although no interaction was found between different groups, it can not be ruled out that the low number of animals may have affected the final results. Furthermore, some more information on materials and methods would be supplied. the authors would specify the number of approval of the ethical committee and some conerns are present on the method of castration: I do not know the legislation in Venezuela, but in other countries it is not acceptable to carry out castration without anesthesia! At least a local anesthesia would be performed, but the authors specified that this was not the case. This creates several concerns about animal welfare.

In any case, the paper does not show any other serious flaws. Some specific comments 

Round 2

Reviewer 1 Report

No more suggestions.

Author Response

There is no more suggestions

Reviewer 3 Report

The authors addressed adequately to the concerns. The limitation regarding the number of animals remains, but the sentence added in the revised version of conclusion is sufficient

Author Response

There is no more suggestions